# Essential roles of Caspase-3 in facilitating Myc-induced genetic instability and carcinogenesis

Ian M Cartwright[1†‡], Xinjian Liu[1†], Min Zhou[1], Fang Li[1], Chuan-Yuan Li[1,2*]

[1]Department of Dermatology, Duke University Medical Center, Durham, United States; [2]Department of Pharmacology and Cancer Biology, Duke University Medical Center, Durham, United States

**Abstract** The mechanism for Myc-induced genetic instability is not well understood. Here we show that sublethal activation of Caspase-3 plays an essential, facilitative role in Myc-induced genomic instability and oncogenic transformation. Overexpression of Myc resulted in increased numbers of chromosome aberrations and γH2AX foci in non-transformed MCF10A human mammary epithelial cells. However, such increases were almost completely eliminated in isogenic cells with *CASP3* gene ablation. Furthermore, we show that endonuclease G, an apoptotic nuclease downstream of Caspase-3, is directly responsible for Myc-induced genetic instability. Genetic ablation of either *CASP3* or *ENDOG* prevented Myc-induced oncogenic transformation of MCF10A cells. Taken together, we believe that Caspase-3 plays a critical, unexpected role in mediating Myc-induced genetic instability and transformation in mammalian cells.

*For correspondence: chuan.li@duke.edu

†These authors contributed equally to this work

Present address: ‡Department of Urology, University of Colorado School of Medicine, Aurora, United States

Competing interests: The authors declare that no competing interests exist.

## Introduction

One of the hallmarks of cancer is increased genomic instability (*Hanahan and Weinberg, 2011*). In addition to DNA replication errors and /or mutations induced by exposure to DNA damaging agents, over-expression of oncogenes have been shown to induce genomic instability (*Bartkova et al., 2005*; *Gorgoulis et al., 2005*). One such oncogene is Myc. As one of the most widely studied oncogenes in cancer biology, it is mutated or over-expressed in multiple types of cancer (*Pelengaris et al., 2002*). Myc is involved in driving cellular proliferation and promoting stem cell self-renewal under normal circumstances. When myc is overexpressed in a cell, it can cause increased genomic instability and promote carcinogenesis (*Karlsson et al., 2003*; *Ray et al., 2006*). Despite numerous studies, the mechanisms involved in myc-induced genomic instability and transformation remain controversial. There are conflicting reports on the mechanism of myc-induced genomic instability and transformation. Several studies suggest that myc-induced genomic instability and carcinogenesis is a result of an overabundance of reactive oxygen species (ROS) (*Vafa et al., 2002*; *Felsher and Bishop, 1999*). However, it has also been reported that myc overexpression can cause DNA damage and transformation in the absence of ROS (*Ray et al., 2006*).

Over-expression of myc has been shown to induce apoptosis (*Evan et al., 1992*; *Harrington et al., 1994*). Until recently, apoptosis has been widely recognized as an anti-carcinogenic process based on the assumption that it is utilized by the host to eliminate damaged cells, including those suffering DNA damage (*Hanahan and Weinberg, 2011*; *Reed, 1999*). However, there has been increasing evidence that apoptosis may in fact been involved in promoting carcinogenesis (*Tang et al., 2012*; *Ichim et al., 2015*; *Liu et al., 2015*). Mammalian cells exposed to external stress can survive activation of the apoptotic cascade and incur increased genetic instability and oncogenic transformation. Studies show that mammalian cells that survive apoptosis experience

**eLife digest** Healthy cells can become cancerous if their DNA is damaged and not repaired properly, leading to changes in the DNA known as mutations. The cells tend to accumulate more and more mutations – a phenomenon known as genomic instability – as they transition into cancer cells. A protein called Myc is known to promote genomic instability and contributes to many types of cancers. However, high levels of Myc proteins in a cell can also activate proteins that trigger a process called apoptosis, which makes the cell commit suicide. This role does not appear to fit with the cancer-promoting properties of Myc because apoptosis is generally thought to protect against cancer by helping to remove damaged cells from the body.

Two of the proteins that Myc activates are known as caspase-3 and endonuclease G. In healthy cells, caspase-3 triggers a series of events that lead to apoptosis, while endonuclease G cuts up DNA in preparation for cell death. However, it is not known how these proteins affect the cancer-promoting properties of Myc.

Here, Cartwright, Liu et al. used a gene editing technique called CRISPR-Cas9 to examine how these apoptosis proteins affect the ability of Myc to promote cancer. Increasing the production of Myc in healthy human mammary cells transformed these cells into breast cancer cells that were capable of forming tumors when injected into mice. However, in cells that could not make caspase-3 or endonuclease G, increasing the production of Myc did not lead to the cells becoming cancerous.

Further experiments show that when Myc levels are high, the activation of caspase-3 and endonuclease G is not sufficient to kill the cells. As a result, endonuclease G causes damage to the DNA and promotes genomic instability. Future studies may focus on understanding exactly how and when the apoptosis proteins can contribute to cancer growth. With this knowledge, it may be possible to prevent or treat Myc-driven cancers by changing how these apoptosis proteins behave.

increased mitochondria membrane permeability (MOMP)(*Ichim et al., 2015*) and sublethal caspase 3 activation (*Liu et al., 2015*), which lead to activation of downstream endonucleases such as CAD and endoG, which in turn cause increased genetic instability and oncogenic transformation.

In the current study, we carried out experiments to examine the potential roles of the cellular apoptotic machinery in Myc-induced mutagenesis and carcinogenesis. We show sublethal activation of caspase 3 and endonuclease G plays an essential role in Myc-induced genetic instability and onco-genic transformation in human cells.

## Results and discussion

Previously, we and others have shown that mammalian cells exposed to external stress such radiation and chemicals could survive the activation of apoptotic caspases. Among the cells that survive caspase activation, elevated DNA damage, such as DNA double strand breaks were observed. In this study, we set out to examine the hypothesis that sublethal activation of apoptotic caspases are involved in Myc-induced genetic instability. Our hypothesis is based on the well-established evidence that Myc over-expression in mammalian cells promotes caspase activation and cell death (*Karlsson et al., 2003*; *Ray et al., 2006*).

To investigate the effects of Myc expression we used a recombinant lentivirus to transduce the c-Myc gene under the control of a constitutively active CMV promoter into MCF10A cells, an immortalized but not transformed human mammary epithelial cell line (*Soule et al., 1990*; *Tait et al., 1990*). We reasoned that if some of the Myc-expressing cells can survive caspase activation, they may possess higher levels genomic instability, similar to those cells that were exposed to ionizing radiation (*Liu et al., 2015*). We quantified Myc's ability to activate Casp3 by immunofluorescence staining of cleaved Casp3 (*Figure 1A,B*). Roughly 6% of Myc-expressing MCF10A cells were observed with having relatively normal nuclei and cleaved caspase 3, as compared to control MCF10A cells where only ~1% of the cells were observed with cleaved caspase 3. We also examined the relationship between Casp3 activation and cellular survival by use of a reporter system described in a prior publication (*Liu et al., 2015*). Our data indicate that irrespective of Casp3 activation status in the presence or absence of Myc expression, 40% or more of the individually sorted MCF10A cells

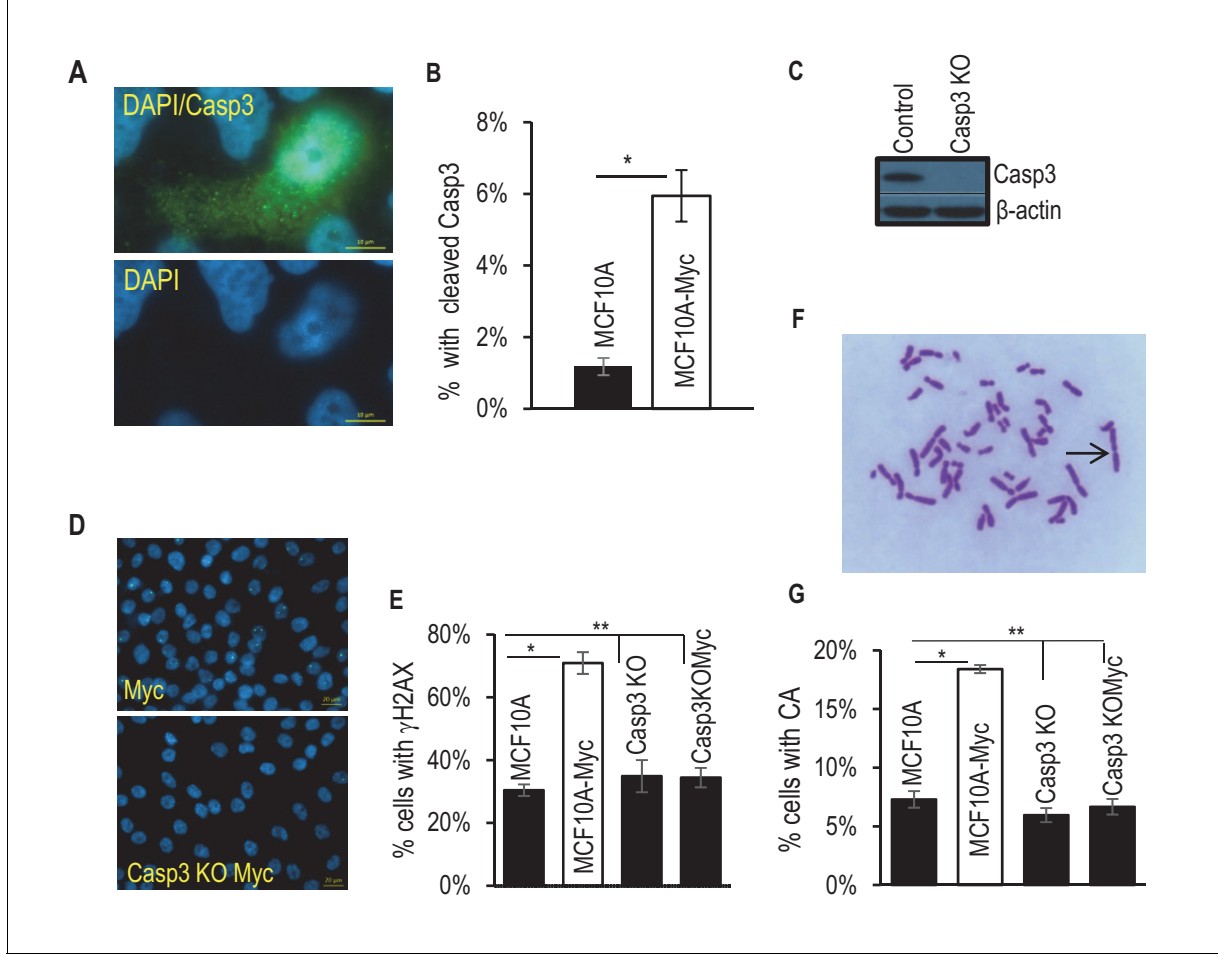

**Figure 1.** Requirement for Casp3 in Myc-induced genomic instability. (**A**) Representative immunofluorescence staining of MCF10A cells with cleaved caspase 3 (green) and DAPI (blue). Scale bar represents 10 μm. (**B**) The proportion of nonapoptotic MCF10A cells presenting with normal nuclear morphology and cleaved caspase 3 signal. (**C**) Western blot analysis of Caspase-3 status in MCF10A cells with or without *CASP3* gene knockout. (**D**) Representative immunofluorescence γH2AX foci (green) and DAPI staining in MCF10A cells with wild type (left panel) and *CASP3KO*. Scale bar represents 20 μm. (**E**) Fraction of cells which stained positive for a γH2AX foci in control and Casp3-deficient MCF10A with or without exogenous expression of Myc, n = 3. (**F**) A representative image of chromosome staining in MCF10A cells. A dicentric chromosome is indicated by an arrow. (**G**) Fraction of cells containing at least one chromosome aberration in control and Casp3-deficient MCF10A with or without exogenous expression of Myc. In B, E, and G, error bars represent standard error of the mean (SEM), * indicates a p value < 0.01, ** indicates a p value > 0.1, Student's t-test, n = 3. For B, E, each data point was derived from the average of three triplicate groups of 150 cells each. In G, each data point was derived from the average of three triplicate groups of 50 cells each.

The following source data and figure supplements are available for figure 1:

**Source data 1.** Data for *Figure 1B*.
**Source data 2.** Data for *Figure 1E*.
**Source data 3.** Data for *Figure 1G*.
**Source data 4.** Data for *Figure 1—figure supplement 3A*.
**Source data 5.** Data for *Figure 1—figure supplement 3B*.
**Source data 6.** Data for *Figure 1—figure supplement 5*.
**Source data 7.** Data for *Figure 1—figure supplement 6B*.

*Figure 1 continued on next page*

*Figure 1 continued*

**Source data 8.** Data for *Figure 1—figure supplement 6C*.

**Source data 9.** Data for *Figure 1—figure supplement 7B*.

**Source data 10.** Data for *Figure 1—figure supplement 7C*.

**Figure supplement 1.** Clonogenic abilities of control and Myc-expressing MCF10A cells with high and low Casp3 reporter (Casp3GFP) activities after being individually sorted into 96-well plates by use of FAC based on their reporter activities.

**Figure supplement 2.** Flow cytometry analysis of annexin v-PE staining in Casp3-GFP transduced MCF10A cells with or without Myc expression.

**Figure supplement 3.** Additional data for the influence of Casp3 on Myc-induced genetic instability in MCF10A cells.

**Figure supplement 4.** Immunofluorescence co-staining of vector control and Myc-transduced MCF10A cells.

**Figure supplement 5.** DNA double-strand repair kinetics in MCF10A cells with various genetic backgrounds.

**Figure supplement 6.** Additional data confirming the role of Casp3 in mediating Myc-induced genomic instability.

**Figure supplement 7.** Role of Casp3 in Myc-induced genetic instability in BJ-1 human fibroblasts.

can form colonies in 96-well plates (*Figure 1—figure supplement 1*). Further flow cytometry analysis showed that despite increased Casp3EGFP activities in Myc-expressing MCF10A cells, the levels of Annexin V staining, a well-recognized marker of apoptosis, did not increase significantly (*Figure 1— figure supplement 2*). Those data provide clear evidence that a significant fraction of MCF10A cells can survive spontaneous or Myc-induced Casp3 activation.

To examine if Casp3 plays a causative role in Myc-induced genomic instability and carcinogenesis, we generated MCF10A and BJ-1hTERT cells with *CASP3* gene knockout by use of the CRISPR/Cas9 technology (*Figure 1C*, *Tables 1–3*). Control and Casp3-deficient MCF10A cells with and without exogenous Myc expression were then analyzed for both chromosome aberrations and γH2AX foci, two well-established markers of genomic instability. In control MCF10A cells, Myc overexpression caused significant increases in both the fraction of cells with and the average numbers per cell of γH2AX foci and chromosomal aberrations (*Figure 1D–G*, *Figure 1—figure supplement 3*). In contrast, Myc overexpression in Casp3-deficient cells induced no increases in the numbers of either chromosome aberrations or γH2AX foci when compared to control MCF10A or *CASP3* knockout (*CASP3KO*) cells without Myc overexpression (*Figure 1D–G*, *Figure 1—figure supplement 3*). To examine the relationship between c-Myc expression and γH2AX foci induction, we carried out immunofluorescence staining of c-Myc and γH2AX (*Figure 1—figure supplement 4*). Our results indicate

**Table 1.** Primary antibodies used in this study.

| Target protein | Antibody source | Clone information |
|---|---|---|
| γH2AX (Ser139) | Upstate Biotechnology | JBW301, Mouse mAb |
| Caspase-3 (full length) | Cell Signaling Technology | 8G10, Rabbit mAb |
| Caspase-3 (cleaved,Asp175) | Cell Signaling Technology | 5A1E, Rabbit mAb |
| EndoG | Chemicon | Rabbit polyclonal |
| HA epitope | Novus Biologicals | Goat polyclonal |
| β-Actin | Novus Biologicals | Mouse mAb |
| c-Myc | Cell Signaling Technology | Rabbit mAb |
| Mito Marker | Thermo Fisher Scientific | N/A |

**Table 2.** Single guided RNA (sgRNA) sequences used in this study.

| Gene | Accession | sgRNA oligo(5'−3')* | Targeted exon |
|------|-----------|---------------------|---------------|
| CASPASE3 | NC_000004 | CACCGcatacatggaagcgaatcaa AAACttgattcgcttccatgtatgC | Exon4 |
| CASPASE3 | NC_000004 | CACCGggaagcgaatcaatggactc AAACgagtccattgattcgcttccC | Exon4 |
| EndoG | NC_000009 | CACCGgggctgggtgcggtcgtcga AAACtcgacgaccgcacccagcccC | Exon1 |
| EndoG | NC_000009 | CACCGcgacttccgcgaggacgact AAACagtcgtcctcgcggaagtcgC | Exon1 |

*Capital letters: enzyme overhangs; non-capital letters: sgRNA target guide sequence.

that c-Myc expression was not homogeneous, perhaps a reflection of the silencing of the CMV promoter that controlled c-Myc expression. Furthermore, γH2AX foci presence did not always correlate with high c-Myc expression, perhaps indicating a stochastic nature of γH2AX foci induction. However, Myc expression did have a strong influence on both the baseline and induced levels of γH2AX foci and their repair kinetics in MCF10A cells. A systematic analysis on the repair of radiation-induced γH2AX foci showed that Myc expression caused not only higher background levels of γH2AX foci but also higher residual foci levels after a significant number of the induced foci were repaired. On the other hand, CASP3 knockout eliminated most of the basal and residual levels of γH2AX foci in MCF10A cells (*Figure 1—figure supplement 5*).

In order to rule out the possibility that our *CASP3KO* cells suffered off-target effects during the generation process, we re-expressed Casp3 in *CASP3KO* MCF10A cells (*Figure 1—figure supplement 6A*) and examined for Myc-induced γH2AX foci. Our results indicate that Casp3 re-expression restored Myc-induced DNA damage foci (*Figure 1—figure supplement 6B*). In a parallel experiment, we expressed a dominant-negative *CAPS3* gene (*dnCASP3*) in Casp3KO cells. DnCasp3 differs from wild-type Casp3 in only a single amino acid that eliminates its cleavage activities (*Stennicke and Salvesen, 1997*). In contrast to wild-type Casp3 re-expression, dnCASP3 re-expression did not restore the ability of Myc to induce γH2AX foci (*Figure 1—figure supplement 6C*).

In order to make sure that our observations so far are not restricted to MCF10A cells, we generated *CASP3* gene knockout cells from hTERT immortalized BJ1 human fibroblast cells (*Figure 1—figure supplement 7A*, *Table 3*) to assess the effects of Casp3 on myc-induced genomic instability. Similar to MCF10A cells, we observed that Myc overexpression resulted in statistically significant increases in γH2AX foci (*Figure 1—figure supplement 7B*) and chromosomal aberrations (*Figure 1—figure supplement 7C*) in cells overexpressing Myc. However, such increases were almost completely eliminated in *CASP3KO* BJ1 cells (*Figure 1—figure supplement 7B,C*), similar to *CASP3KO* MCF10A cells.

Since Myc-induced genomic instability is intimately associated with its ability to transform mammalian cells, we investigated Myc-induced tumorigenicity in MCF10A cells. We initially evaluated

**Table 3.** Mutations at target sequences in various CRISPR knockout MCF10A and BJ-1 hTERT cells.

| | | 5'......3' | Mutation |
|---|---|-----------|----------|
| Casp3 KO | MCF10A | Clone1: AAAGATCATACATGGAAGCGAATCAATGGA - - - - - - - ATAT<br>Casp3: AAAGATCATACATGGAAGCGAATCAA<mark>TGG</mark>ACTCTGGAATAT | 7 bp deletion |
| Casp3 KO | BJ-1hTERT | Clone28: AAAGATCATACATGGAAGCGAATCAATG - - - deletion——————<br>Casp3: AAAGATCATACATGGAAGCGAATCAA<mark>TGG</mark>ACTCTGGAATAT | 193 bp deletion |
| EndoG KO | MCF10A | Clone13: —————— deletion——————<br>EndoG: TGCCA<mark>CCA</mark>ACGCCGACTACCGCGGCAGTGGCTTCGACCGCG | 169 bp deletion |

Note: Red: sgRNA sequence; Yellow: PAM sequence; Bold: insertion sequence; -: deletion sequence. In all cases, knockout clones that showed both clear absence of target protein expression and gene mutations were chosen. In addition, in most cases, only those clones with homozygous mutations (where both copies of the gene showed the same mutation) were chosen for convenience.

Myc-induced tumorigenicity of MCF10A cells by use of the soft agar colony forming assay, a well-establish assay that evaluates the anchorage independence ability of putative tumor cells. Our results indicate that Myc overexpression in control MCF10A resulted in a significant increase in the number of observed soft agar colonies (*Figure 2A,B*). However, such increases were completely absent in Casp3 knockout cells (*Figure 2A,B*). The causative role for Casp3 in this process was further demonstrated in Casp3KO cells with Casp3 re-expression, the ability of Myc to induce soft agar colony formation was restored (*Figure 2C*). In control MCF10A cells, expression of an exogenous Casp3 caused no increase in Myc-induced soft agar colony formation (*Figure 2C*).

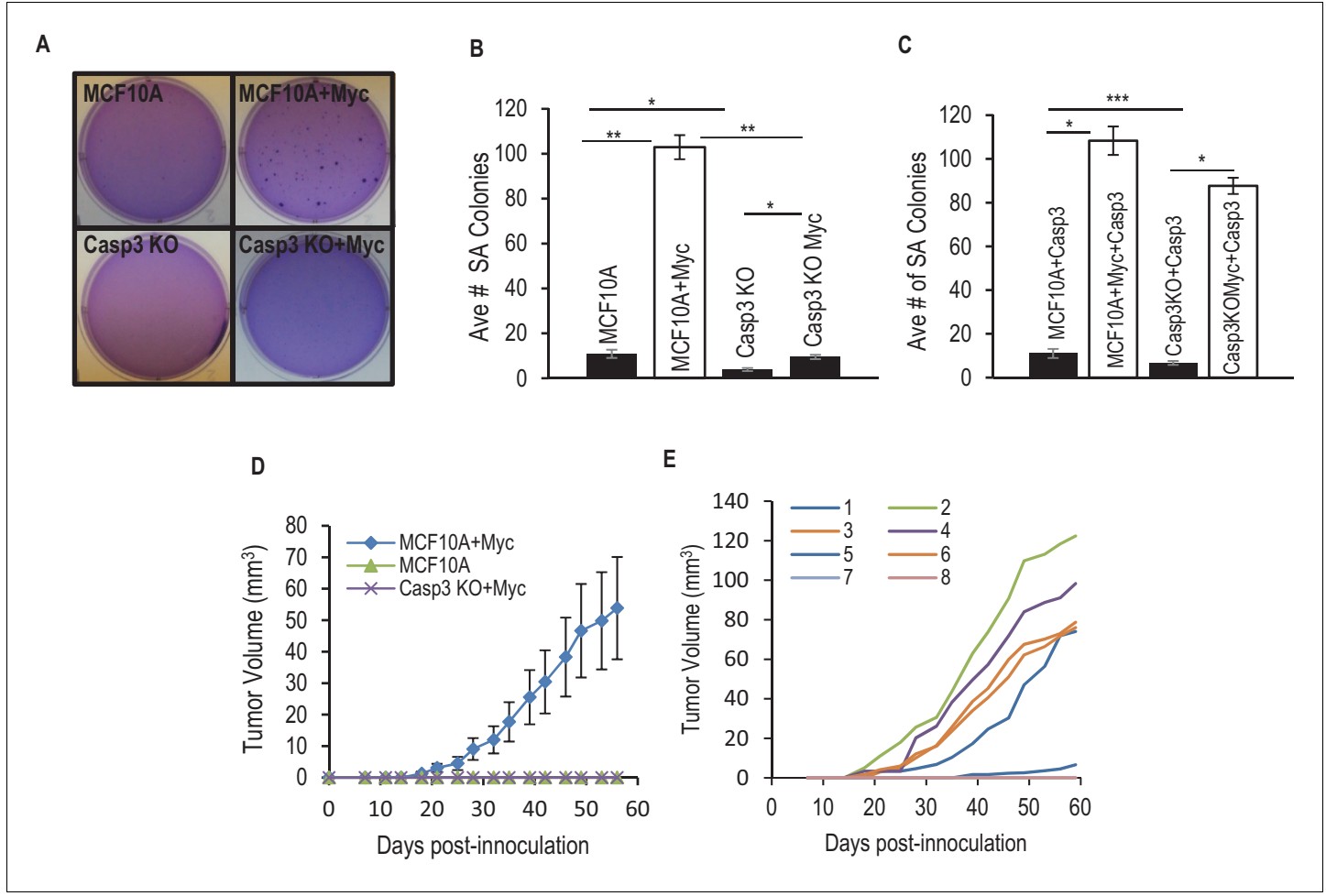

**Figure 2.** Requirement for Casp3 in Myc-induced transformation. (A) Depicts colonies which grew in soft agar. (B) Average number of colonies in soft agar in control and Casp3-deficient cells with or without Myc expression. (C) Average number of colonies in soft agar in wild type and Casp3 knockout MCF10A cells with Casp3 re-expression in the absence or presence of Myc over-expression. (D) Tumor growth from control, Myc-overexpressing, and Casp3KO cells with Myc over-expressiong in nude mice. (E) Individual tumor sizes in nude mice form wild-type cells with Myc over-expression. The error bars in B, D, and E represent standard error of the mean (SEM). * Indicates p value < 0.001, ** indicates p value << 1e$^{-5}$, *** indicates a p value > 0.1. Student's t-test was used to calculate the p-values in B and C. N = 3 for B and C. N = 5 for D.

The following source data is available for figure 2:

**Source data 1.** Data for *Figure 2B*.
**Source data 2.** Data for *Figure 2C*.
**Source data 3.** Data for *Figure 2D&E*.

In a further experiment, we examined the ability of Myc-transduced control or *CASP3KO* cells to form tumors in nude mice. Despite injection of $4 \times 10^6$ cells each, only the group of nude mice that were injected with Myc-transduced control MCF10A cell were able to form tumors after 8 weeks of observation (*Figure 2D*), thereby confirming that both Myc over-expression and an intact *CAPSP3* gene are required for oncogenic transformation. A more detailed examination showed that in Myc over-expressing MCF10A cells, five out of eight injected sites formed tumors, albeit with varying growth kinetics (*Figure 2E*). The in vivo data here are striking in that Casp3 deficiency completely blocked the ability of Myc to transform MCF10A cells. In vitro, although Casp3 deficiency significantly reduced Myc-induced soft agar colony formation, it was not completely blocked. The discrepancy between the in vitro and in vivo assays perhaps reflected the different properties that the two assays are measuring.

We next sought to determine the downstream effectors of Casp3 in mediating Myc-induced genomic instability and oncogenic transformation. In a previous study, endonuclease G (endo G), an apoptotic endonuclease that normally resides within the mitochondria and migrates to the nucleus after activation of apoptotic cascade (*Parrish et al., 2001*; *Li et al., 2001*), is responsible for much of the Casp3-induced genomic instability after stress exposure. In order to evaluate if endoG is involved in Myc-induced genomic stability, we determined the cellular location of endoG in control and Myc expressing MCF10A cells by use of immunofluorescence staining. Our results show that there was a significant increase in the fraction of cells with endoG nuclear migration in Myc-expressing MCF10A vs control cells (11% vs 2%, *Figure 3A,B*). However, the increase was completely eliminated in *CASP3* knockout cells (*Figure 3B*).

In order to determine if endoG plays a causative role in Myc-induced genomic damage and transformation, we created MCF10A cells with *ENDOG* gene knockout by use of CRISPR/Cas9 (*Figure 3C*). We then evaluated the abilities of Myc to induce γH2AX foci and oncogenic transformation. Our results show that ENDOG deletion was able to eliminate both Myc-induced γH2AX foci (*Figure 3D* and *Figure 3—figure supplement 1*) and oncogenic transformation as evaluated by use of the soft agar colony-forming assay (*Figure 3E*).

To further determine the relationships among Myc expression, CASP3 status, ENDOG status, and apoptosis, we carried out flow cytometry analysis of Annexin V and PI (propidium iondine) staining, which allowed for the identification of different stages of cellular apoptosis. Our results (*Figure 3—figure supplement 2*) indicate that CASP3KO caused small reductions in both Annexin V+ and Annexin V+/PI+ MCF10A cells when compared with control cells. However, ENDOG knockout caused small increases in fractions of Annexin V+/PI- and Annexin V-/PI+ cells when compared with control cells. Myc expression, on the other hand, caused increases in fractions of Annexin V+/PI+ cells I all three cell populations. Overall, while the relative changes could be sizable, the absolute changes caused by the knockouts or Myc expression in terms of PI+ or Annexin V+ cells were small.

Increased ROS has been previously implicated in Myc-induced carcinogenesis. Our data so far has suggested strongly that the Casp3 activation and endoG release from the mitochondria played decisive roles in Myc-induced carcinogenesis. In order to determine if ROS levels in MCF10A cells correspond with oncogenic transformation, we did DCFDA-based flow cytometry analysis ROS levels in various MCF10A-derived cells (*Figure 3—figure supplement 3*). Our results indicate that CASP3 or ENDOG knockout caused small increase and decrease in MCF10A cells (*Figure 3—figure supplement 3*, top panels), respectively. Forced MYC expression increased ROS in wild-type MCF10A cells but not in CASP3KO or ENDOGKO MCF10A cells (*Figure 3—figure supplement 3*, mid-panels). Further, Myc was able to induce significantly more ROS in wild-type MCF10A cells than CASP3KO cells, but about equal levels of ROS in wild type vs ENDOGKO cells (*Figure 3—figure supplement 3*, lower panels). Those data suggest that although ROS production appeared to track with carcinogenesis in wild-type and CASP3KO cells, it did not in ENDOGKO cells.

In order to determine whether Casp3 activation and endoG release is the result of partial damage of many mitochondria vs severe damage to a small number of mitochondria, a quantitative PCR (qPCR) analysis of cytoplasmic mitochondrial DNA (mtDNA) levels was done (*Figure 3—figure supplement 4*). Our results indicate that MYC expression in MCF10A cells caused a significant increase in cytoplasmic mtDNA levels, indicating that a significant fraction of mitochondria had compromised membrane integrity. On the other hand, CASP3KO significantly reduced cytoplasmic mtDNA levels in MCF10A cells with or without Myc over expression. While we are not clear of the exact cause of

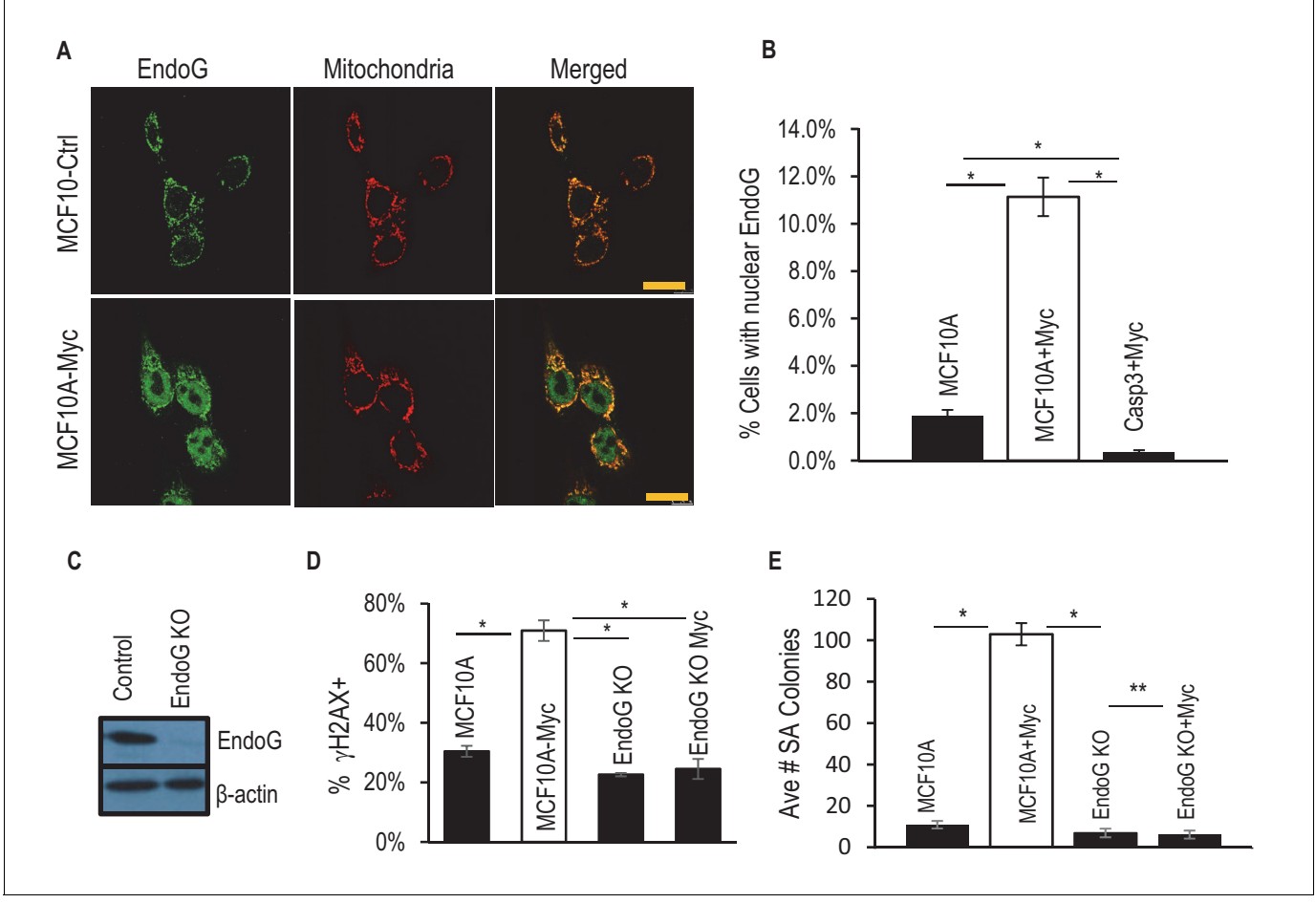

**Figure 3.** Requirement of EndoG in Myc-induced genomic instability and transformation. (**A**) Immunofluorescence staining of of MCF10A with antibodies to EndoG (green), mitochondria (Orange) and DAPI (blue). Scale bar represents 20 μm. (**B**) Fraction of MCF10A cells with activated EndoG (EndoG signal within the nucleus). Error bar indicates SEM. (**C**) Western blot analysis fo EndoG expression in wild type or endoG knockout MCF10A cells. (**D**) Fraction of cells which stained positive for a γH2AX foci in control and EndoG-deficient MCF10A with or without exogenous expression of Myc. (**E**) Influence of endoG status on Myc-induced transformation of MCF10A cells, as indicted by soft agar (SA) colony formation. * Indicates p value < 0.001, **p>0.05. Student's t-test in B, D, E. Error bars represent SEM. In B, D, each data point was derived from the average of three triplicate groups of 150 cells each. In E, n = 3.

The following source data and figure supplements are available for figure 3:

**Source data 1.** Data for *Figure 3B*.
**Source data 2.** Data for *Figure 3D*.
**Source data 3.** Data for *Figure 3E*.
**Source data 4.** Data for *Figure 3—figure supplement 1*.
**Source data 5.** Data for *Figure 3—figure supplement 4*.
**Figure supplement 1.** Additional data showing the average number of γH2AX foci per cell in control and EndoG deficient MCF10A cells with or without exogenous expression of Myc.
**Figure supplement 2.** Flow cytometry analysis of various MCF10A cells for PI and annexin V staining to quantitate the fraction of cell undergoing early (lower right quadrants), late (top right quadrants) apoptosis, and necrosis (top left quadrants).
**Figure supplement 3.** Flow cytometry level analysis of reactive oxygen species (ROS) levels in various MCF10A cells.
*Figure 3 continued on next page*

*Figure 3 continued*

**Figure supplement 4.** Relative levels of mitochondrial DNA that leaked into the cytoplasm in control as wells genetically modified MCF10A cells.

this, we speculate that Casp3 and other upstream factors that promote mitochondrial leakage form a positive loop to promote mitochondrial leakage with or without Myc expression.

To further examine if endoG leakage from the mitochondria and migration is sufficient to cause genomic instability and oncogenic transformation, we engineered a modified endoG protein where the native mitochondrial localization signal is switched to a nuclear localization signal (NLS-EndoG, *Figure 4A*) and transduced it into MCF10A*CASP3KO* cells with or without Myc expression (*Figure 4B*). Immunofluorescence staining confirmed the nuclear localization of the engineered endoG (*Figure 3C*). We then determined the incidence of γH2AX foci in transduced cells. Our results indicate the NLS-endoG expression restored the depleted γH2AX foci induction by Myc in *CASO3KO* cells (*Figure 4D*). In fact, NLS-endoG alone was sufficient to induce γH2X foci in *CASP3KO* cells to levels induced by Myc (*Figure 4D*).

We next examined the influence of NLS-EndoG on oncogenic transformation by use of the soft agar assay. Our results show that NLS-EndoG restored the ability of Myc to induce oncogenic transformation in Casp3 knockout MCF10A cells (*Figure 4E*). However, NLS-EndoG expression alone was not sufficient to induce oncogenic transformation despite its ability to induce DNA damage foci to levels similar to Myc expression (*Figure 4E*). Those results suggest while endoG nuclear migration is a necessary condition for Myc-induced oncogenic transformation, it is not sufficient by itself. Additional activities of Myc are clearly required in the transformation process.

A further tumor formation experiment was conducted in nude mice to examine the role of NLS-EndoG in oncogenic transformation. NLS-EndoG restored the ability of Myc to induce oncogenic transformation in *CASP3KO* MCF10A cells (*Figure 4F*), with five out of eight inoculations formed tumors. On the other hand, NLS-EndoG alone was not able to make MCF10A cells with *CASP3* gene knockout to become tumorigenic, consistent with results from soft agar colony formation assay (*Figure 4E*).

Despite being one of the first oncogenes identified and having numerous studies dedicated to discovering its roles in cancer biology, there are still many gaps in our knowledge about Myc. The present study provides significant new insights into the roles of Myc in carcinogenesis. In particular, it resolves two apparently paradoxical observations regarding Myc: its ability to stimulate apoptosis and to induce genomic instability and oncogenic transformation. In many instances, such as in the case of p53, apoptosis induction is anti-carcinogenic due to its ability to remove damaged cells from the body. However, in the case of Myc, the apoptotic machinery, is exploited by Myc as a vehicle to cause genomic instability and induce oncogenic transformation. Most strikingly, our data suggest that Casp3 and Endo G, two well-established apoptosis effectors, are activated and required for Myc-induced oncogenic transformation. This finding suggest that Myc-induced activation of apoptosis, instead of being a result of Myc induced cellular stress, is actually part and parcel of Myc's capacity to induce mammalian transformation.

One important piece of information that remain missing is how Myc induces mitochondrial leakage and Casp3 activation. While we do not have any experimental evidence at present, it is possible that deregulation in mitochondrial biogenesis, which is known to be stimulated by Myc, may be responsible for it, as suggested previously (*Dang et al., 2005*; *Zhang et al., 2007*; *Ahuja et al., 2010*). This possibility should be investigated in future studies since it may holds the key to Myc's powerful oncogenic abilities.

On the surface, our discovery appears to be contrary to the established paradigm that apoptosis is a key barrier for carcinogenesis (*Hanahan and Weinberg, 2011*). However, it is consistent with an increasing body of literature that suggest a pro-oncogenic role for apoptosis and some apoptotic factors (*Tang et al., 2012*; *Ichim et al., 2015*; *Liu et al., 2015*). The key conceptual advance in the studies is the realization of that cells exposed to stress can survive caspase activation (*Tang et al., 2012*; *Ichim et al., 2015*; *Liu et al., 2015*; *Ding et al., 2016*; *Ichim and Tait, 2016*). The observation of such survival in development (*Ding et al., 2016*), chemical exposure (*Tang et al., 2012*;

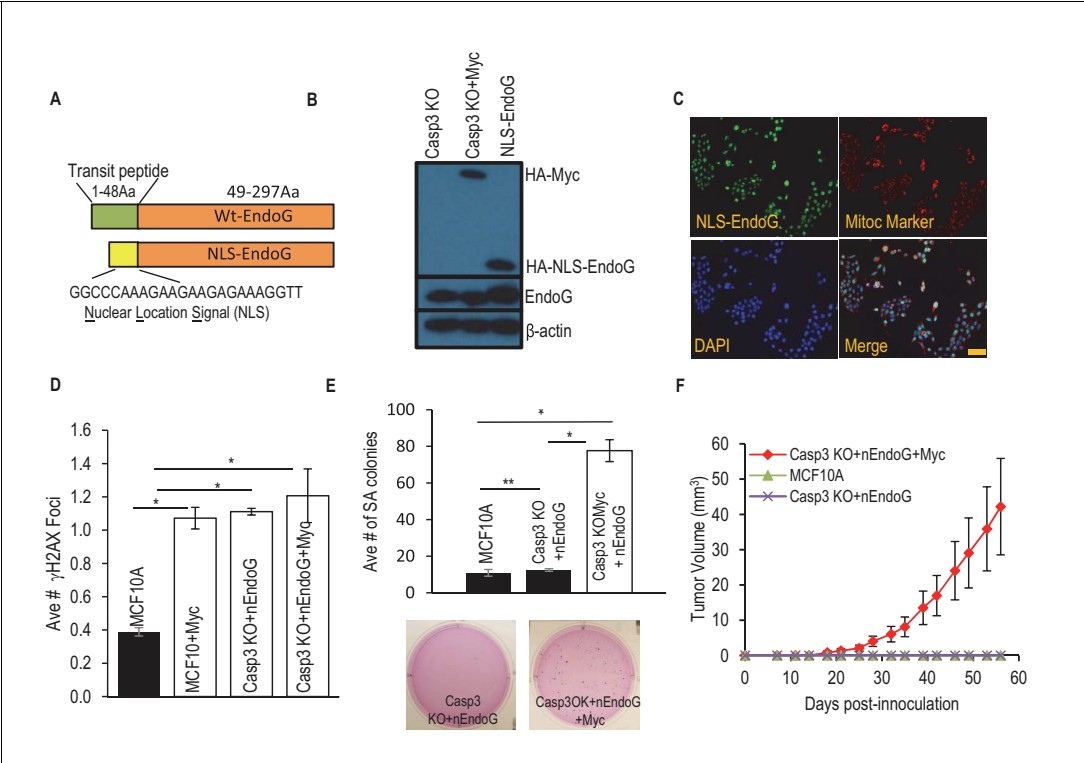

**Figure 4.** A nucleus-located EndoG restores Myc induced transformation in Casp3-deficient cells. (**A**) A diagram showing a re-engineered endoG with a nucleus localization signal (NLS) at its tagged N-terminal. (**B**) Western blot demonstrating exogenous myc and NLS-EndoG expression by use of an anti-HA antibody. (**C**) Immunoflouresence staining of EndoG (green), mito marker (red), and DAPI (blue) in NLS-EndoG transduced *CASP3KO* cells. Scale bar indicates 20 µm. (**D**) Average number of γH2AX foci per cell in MCF10A with or without nEndoG in the absence or presence of Myc or Casp3KO. (**E**) The influence of nEndoG on soft agar formation in MCF10A-Casp3KO cells with or without Myc gene expression. (**F**) Xenograft tumor formation in nude mice from MCF10A-Casp3KO cells with NLS-EndoG (nEndoG) and/or Myc expression. In D, E, F error bars represent standard error of the mean (SEM), * indicates p value < 0.001 while ** Indicates p value > 0.1. Student's t-test. In D, each data point was derived from the average of three triplicate groups of 150 cells each. In E, n = 3. In F, n = 5.

The following source data is available for figure 4:

**Source data 1.** Data for *Figure 4D*.
**Source data 2.** Data for *Figure 4E*.
**Source data 3.** Data for *Figure 4F*.

*Ichim et al., 2015*), and radiation exposure (*Liu et al., 2015*) indicate that it is a wide spread phenomenon. Our observation of cells surviving Myc-induced caspase activation is a significant extension of those earlier observations and may provide important insights into the relationship between apoptosis and oncogene-induced transformation beyond that of Myc.

Taken together, our study provides crucial evidence that Casp3 and endo G, two key factors in the canonical apoptosis pathway, play essential, facilitative roles in Myc-induced oncogenic transformation.

# Materials and methods

## Cell lines and tissue culture

Early passage, immortalized, non-transformed human breast epithelial cell line, MCF10A, was a kind gift from Dr. Hatsumi Nagasawa of Colorado State University (Fort Collins, CO). MCF10A growth

medium was composed of Dulbacco's modified Eagle's medium (DMEM)/F12 (Sigma) supplemented with 5% donor horse serum (Sigma), 20 ng/ml of epidermal growth factor (EGF; R&D Systems), 0.5 µg/ml hydrocortisone (Sigma), 100 ng/ml cholera toxin (Sigma), 10 µg/ml insulin (Invitrogen), and 100 units/ml penicillin and 100 µg/ml streptomycin. hTERT immortalized, non-transformed human fibroblast cell line, hTERT BJ-1, was a kind gift from Dr. Takamitsu Kato of Colorado State University. hTERT BJ-1 growth medium was composed of DMEM supplemented with 10% fetal bovine serum (Sigma) and 100 units/ml penicillin and 100 µg/ml streptomycin. The identities of both MCF10A and hTERT-BJ1 were authenticated through STR profiling methods. Throughout the course of the experiments, the cells were also checked periodically for the absence of mycoplasma contamination.

### γH2AX foci assay

For γH2AX foci assays, the cells were synchronized in G1 using the well-established double-thymidine block protocol (*Bostock et al., 1971*). Briefly, cells were plated on glass-bottom 35 mm petri dishes (MatTek, Ashland, MA) and cultured with growth medium for overnight. They were incubated with 2 mM thymidine for 18 hr, washed 2x with PBS, and incubated for 10–12 hr in growth media. They were then incubated for 15–18 hr with 2 mM thymidine. After synchronization, cells were fixed with 4% PFA and permeabilized and blocked in PBS containing 0.1% Triton X-100, 5% goat serum, and 1% BSA. Cells were incubated with a primary antibody against γH2AX (Upstate Biotechnology, Lake Placid, NY), wash with PBS and incubated with a secondary antibody conjugated with Alexa Fluor 488 (Invitrogen). Cells were mounted with mounting medium (Vector Laboratories) containing DAPI. Fluorescent images of γH2AX were acquired with a Zeiss fluorescence microscope with a 63x oil objective (Axio Observer Z1). For each experimental group we observed 150 cells in triplicate.

### Chromosome aberration analysis

We carried out chromosome aberration analysis in cultured cells following an established protocol (*Savage, 1976*). We analyzed for various chromosome/chromatid aberrations that include breaks/gaps, dicentrics, centric/acentric rings, and translocations. Each data points represent data from 50 cells in triplicate.

### CRISPR/Cas9-mediated gene knockout and lentivirus production

We made various cells deficient in various genes by use of the CRISPR/Cas9 technology. Single-guided RNA (sgRNA) sequences targeting the genes were generated with the use of a free online CRISPR design tool (crispr.mit.edu). The sgRNA sequences used were listed in *Table 2*. Annealed double stranded sgRNA oligos were ligated into the lentiCRISPR vector (*Cong et al., 2013*) (deposited by Dr. Feng Zhang to Addgene, Cambridge, MA) at BsmBl site, which co-express cas9 and sgRNA in the same vector. The constructed CRISPR lentivirus vectors were then packaged according to a standard protocol. To produce lentiviral vectors, lentiviral plasmids with the target genes were transduced into 293 T cells together with second-generation packaging plasmids (psPAX2, pMD2.G) following previously published procedures: http://tronolab.epfl.ch/lentivectors.

### Immunofluorescence staining

Cells were cultured on glass-bottom 35 mm petri dishes. Cells were fixed with 4% paraformaldehyde (PFA) in PBS for 15 min, permeabilized and blocked with PBS containing 5% goat serum, 0.1% Triton X-100, and 1% bovine serum albumin (BSA) for 45 min. Fixed cells were incubated with primary antibodies for cleaved Caspase-3, γH2AX, or EndoG overnight at 4C, followed by incubation with appropriate Alexa Fluor 488-conjugated secondary antibodies (Invitrogen, Carlsbad, CA) for 1 hr and mounted with mounting medium (Vector Laboratories, CA) containing DAPI. Fluorescent images were acquired with a Zeiss fluorescence microscope with a 63x oil objective (Axio Observer Z1).

### Soft-agar assay

The soft-agar assay was carried out following an established procedure (*Cifone and Fidler, 1980*). About $1 \times 10^4$ MCF10A cells in growth medium were plated into six-well plates with 1.5 ml 0.3% (m/v) low-melting agar (BD, Sparks, MD, which was overlaid onto 1.5 ml 0.6% (w/v) bottom agar layer. Soft-agar colonies were maintained at 37°C and fed twice a week by drop-wise addition of

growth medium for colony formation. After 21 days in culture, the colonies were counted after staining with 0.005% crystal violet.

## Flow cytometry-based analysis of ROS

In order to measure reactive oxygen species, the cells were labeled with DCFDA (20 µM) according to the manufacturer's instruction that comes with the ROS kit (Abcam, Cambridge, MA). The cells were then analyzed by use of flow cytometry.

## Q-PCR analysis of mtDNA

To measure cytoplasmic mtDNA in various MCF10A-derived cells, the cytoplasmic fraction of the cellular lysates were isolated according to a published protocol (*Bronner and O'Riordan, 2016*). To quantify mtDNA, a segment of the mtND5 gene was amplified by use of the primer pair (*Neufeld-Cohen et al., 2016*): forward 5'-ACGAAAATGACCCAGACCTC-3', rev 5'-GAGATGACAAATCC TGCAAAGATG-3' through Q-PCR. A pair of primers for 18 s rDNA (Forward: 5'-TAGAGGGACAAG TGGCGTTC-3' Reverse: 5'-CGCTGAGCCAGTCAGTGT-3') was used as control.

## Tumor formation in nude mice

Animal experiments conducted in this study were approved by the Duke University Institutional Animal Use and Care Committee (protocol# A195-14-08). To confirm the tumorigenicity of myc overexpressing MCF10A cells, about $4 \times 10^6$ cells were injected subcutaneously into the flanks of 6–8 week-old, female athymic nude mice (Jackson Laboratories) in 50 µl of 1:1 Matrigel (Corning) and PBS. After inoculation, the growth of tumors was evaluated once a week for 8 weeks.

## Statistical analysis

Student's t-test was used to evaluate the significance of differences between different experimental groups. In most cases, a p-value of less than 0.05 was deemed as significant while a p-value of more than 0.05 was deemed not significant.

# Acknowledgements

We thank Dr. Feng Zhang (MIT) for making their CRISPR/Cas9 plasmids available through Addgene (Cambridge, MA). We also thank Drs. Hatsumi Nagasawa and Takamitsu Kato for sending us MCF10A and hTERT-BJ1 cells, respectively. We thank the Flow Cytometry Core Facility at Duke Cancer Institute for providing expert FACS analysis and sorting. The authors declare no conflicts of interest for the present work. This study was supported in part by grants CA155270, ES024015, and CA2008852 from the National Institutes of Health, the Duke Skin Disease Research Core Center grant (NIAMS-AR066527) (C Li); and NSFC81572788 (X Liu) from the National Science Foundation of China.

# Additional information

## Funding

| Funder | Grant reference number | Author |
|---|---|---|
| National Cancer Institute | CA155720 | Chuan-Yuan Li |
| National Institute of Environmental Health Sciences | ES024015 | Chuan-Yuan Li |
| National Institute of Arthritis and Musculoskeletal and Skin Diseases | AR066527 | Chuan-Yuan Li |
| National Cancer Institute | 2008852 | Chuan-Yuan Li |

The funders had no role in study design, data collection and interpretation, or the decision to submit the work for publication.

## Author contributions
IMC, Conceptualization, Data curation, Software, Formal analysis, Validation, Investigation, Methodology, Writing—original draft, Writing—review and editing; XL, Data curation, Validation, Investigation, Methodology, Writing—review and editing; MZ, Resources, Validation, Investigation, Methodology; FL, Supervision, Investigation, Methodology; C-YL, Conceptualization, Data curation, Funding acquisition, Project administration, Writing—review and editing

## Author ORCIDs
Chuan-Yuan Li, http://orcid.org/0000-0002-0418-6231

## Ethics
Animal experimentation: The animal experiments conducted in this study were approved by the Duke University Institutional Animal Use and Care Committee (IACUC) with the protocol number A195-14-98.

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
