## [Decision Letter]

Thank you for submitting your article "Essential roles of Caspase-3 in facilitating Myc-induced genetic instability and carcinogenesis" for consideration by *eLife*. Your article has been favorably evaluated by Kevin Struhl (Senior Editor) and three reviewers, one of whom, Chi Van Dang, is a member of our Board of Reviewing Editors. The following individual involved in review of your submission has agreed to reveal their identity: Dean Felsher (Reviewer #2).

The reviewers have discussed the reviews with one another and the Reviewing Editor has drafted this decision to help you prepare a revised submission.

Summary:

The manuscript by Cartwright et al. reports data which are consistent with a previously unsuspected role of MYC-induced pre-apoptotic release of caspase-3 and the downstream endonuclease ENDOG in mediating genetic instability and contributing to transformation of MCF10A cells. They perform studies in MCF10A and TERT-immortalized human fibroblasts and measure genomic instability including measures of γH2AX and chromosomal instability and in vitro transformation and in vivo growth in nude mice. Specifically, deletion of CASP3 or ENDOG blocked MYC-mediated genomic instability and neoplastic transformation. The authors used straightforward experimental approaches with γH2AX as a marker of DNA damage and CRISPR/Cas9 mediated knockout of CASP3 or ENDOG in two cell systems to test their hypotheses. They also measured chromosomal abnormalities by karyotyping. They demonstrate the nuclear presence of EndoG was higher in MYC-overexpressing cells than control and this was diminished with CASP3 knockout. To determine whether EndoG is responsible for DNA damage, they knocked out ENDOG and found a reduction in DNA damage in MYC-overexpressing cells. This was negated by reintroduction of a nuclear-targeted EndoG (NLS-linked) transfected back into EndoG knockout cells. Nuclear-targeted EndoG, while sufficient to increase γH2AX staining and necessary for MYC-mediated transformation, is insufficient for neoplastic transformation. Nuclear-targeted EndoG, however, could rescue MYC-mediated transformation and tumorigenesis in MYC-overexpressing MCF10A with CASP3 knocked out. Overall, the authors provide compelling evidence for a MYC-mediate near-death cellular state in which leakage of CASP3 could activate ENDOG to cause DNA damage and contribute to neoplastic transformation.

Essential revisions:

1) The use of additional quantitative assays would greatly bolster this study:a) The authors should provide quantitative assessment of apoptosis using Annexin V + PI staining and flow cytometry in the different cell lines at baseline. These simple experiments will help bolster these intriguing novel observations.

b) DNA break repair assay.

c) ROS production. While the rescue experiments, particularly by nuclear-targeted EndoG in CASP3-knockout cells, seems to pinpoint the pathway from MYC to DNA damage, MYC can also induce ROS production. As such, documentation of baseline ROS in the different cell lines using flow cytometry should address whether ROS tracks with tumorigenesis.

d) Cytoplasmic mtDNA levels (by PCR relative to genomic DNA) to determine the extent of mitochondrial damage in MYC-overexpressing cells. This would address whether release of Casp3 is from partially damaged mitochondria or during fission and fusion (induced by MYC as reported in the literature) or due to small percentages of mitochondria that are full damaged.

2) In Figure 1, the percentage of cells with cleaved Caspase-3 in Myc overpressed cells is about 6%, which is about 4-5 times higher than in control cells. The authors should document the extent of MYC expression among the MCF10A cells by anti-MYC antibody immunofluorescent microscopy to accompany the γH2AX staining. Presumably, MYC staining should be quite uniform.

3) Figure 1—figure supplement 1: there seems to be no difference in cell survival in either Casp3GFP high or low sorted cells. The authors should document by Annexin V (or other apoptotic markers) staining to verify whether apoptosis process is initiated in Casp3GFP high vs. low cells.

4) Do the authors find that knocking out Caspase 3 reduced the frequency of tumorigenic cells or is there an absolute prevention of tumorigenic growth? This aspect should be discussed. Further Caspase 3 may have other functions that are required for sustained tumor growth, such as suppressing MYC-mediated apoptosis. The authors need to acknowledge this possibility.

5) The authors did not define the mechanism by which overexpressed MYC induces low level release of Casp3. This should be discussed to encourage further studies regarding MYC's effects on deregulated mitochondrial biogenesis and function.

---

## [Author Response]

*Essential revisions:*

*1) The use of additional quantitative assays would greatly bolster this study:a) The authors should provide quantitative assessment of apoptosis using Annexin V + PI staining and flow cytometry in the different cell lines at baseline. These simple experiments will help bolster these intriguing novel observations.*

We agree with this suggestion and carried out flow cytometry analysis of Annexin V +PI staining of various cell lines used in our study. The data, now shown as Figure 3—figure supplement 2, indicate the Myc over expression caused only a small but clear increase in AnnexinV and PI double positive fraction in parental MCF10A cells. The same is true for both Casp3KO or EndoGKO MCF10A cells. The fractions of AnnexinV+PI‐ cells were slightly higher in EndoG knockout cells but lower in Casp3 knockout cells when compared with parental MCF10A cells.

b) DNA break repair assay.

In order to determine the capacity of wild type and gene knockout MCF10A cells to repair DNA double strand breaks, we carried out experiments to measure the kinetics of x‐ray induced γH2AX foci induction in MCF10A and MCF10ACasp3KO cells with or without Myc gene expression (Figure 1—figure supplement 5). Our data indicated that Myc over‐expressing wild type MCF10A cells had the highest fraction of cells with γH2AX foci in un‐irradiated cells, as expected. After 3 Gy of irradiation, all cell types incurred significant DNA damage with γH2AX foci with Myc‐expressing wild type MCF10A cells reaching over 98% of cells that were γH2AX positive while the other cells reaching close to 90%. In all cells, a significant fraction of the induced γH2AX foci were gone by 60 min after irradiation. However, there were significant fractions of residual DNA damage that were not repaired in all irradiated cells, with Myc expression MCF10A cells having over 75% cells that were γH2AX+ while the other cells having 35‐45% fraction that were γH2AX+.

*c) ROS production. While the rescue experiments, particularly by nuclear-targeted EndoG in CASP3-knockout cells, seems to pinpoint the pathway from MYC to DNA damage, MYC can also induce ROS production. As such, documentation of baseline ROS in the different cell lines using flow cytometry should address whether ROS tracks with tumorigenesis.*

We agree that it is important to examine the roles of ROS in MYC induced carcinogenesis. Therefore, we did a systematic analysis of ROS levels in various MCF10‐derived cell lines by use of the DCFDA ROS staining assay and flow cytometry analysis (Figure 3—figure supplement 3). Our data indicate that CASP3 knockout reduced ROS levels (top left panel) while ENDOG knockout increased ROS levels (top right panel). On the other hand in MYC overexpression increased ROS levels significantly in MCF10A cells while such increases were eliminated in both CASP3KO cells and ENDOGKO cells (mid‐panels). It is also important to point out that Myc‐expressing ENDOG knockout cells had almost the same level of ROS levels as Mycexpressing wild type MCF10A cells (lower right panel). Therefore, ROS appears to track with tumorigenesis in Myc‐expressing MCF10A cells but not with baseline ROS levels in ENDOG knockout cells or Myc‐ENDOG knockout cells. Furthermore, the fact that nuclear endoG (NLSendoG) can compensate for the loss of Casp3 indicate that nuclease‐mediated DNA damage is the dominant mechanism in Myc‐induced carcinogenesis over ROS.

*d) Cytoplasmic mtDNA levels (by PCR relative to genomic DNA) to determine the extent of mitochondrial damage in MYC-overexpressing cells. This would address whether release of Casp3 is from partially damaged mitochondria or during fission and fusion (induced by MYC as reported in the literature) or due to small percentages of mitochondria that are full damaged.*

We agree with the above assessment and carried out quantitative PCR analysis of an mtDNA encoded gene mtND5 to measure the levels of cytoplasmic mtDNA levels in various MCF10A‐derived cells. Our results (Figure 3—figure supplement 4) indicate that Myc expression significantly increased the levels of cytoplasmic mtDNA. We believe this points to a general increase in the number of damaged mitochondria instead of a total damage in a small number of mitochondria as the latter would not have increased cytoplasmic mtDNA significantly. Interestingly, Casp3 knockout significantly reduced the levels of cytoplasmic mtDNA in baseline as well as Myc‐expressing MCF10A cells, suggesting that mitochondrial leakage and Caspase activation forms a feedback loop.

*2) In Figure 1, the percentage of cells with cleaved Caspase-3 in Myc overpressed cells is about 6%, which is about 4-5 times higher than in control cells. The authors should document the extent of MYC expression among the MCF10A cells by anti-MYC antibody immunofluorescent microscopy to accompany the γ-H2AX staining. Presumably, MYC staining should be quite uniform.*

We followed the reviewers’ suggestion carried out immunofluorescence staining of Myc and γH2AX in MCF10A‐Myc cells (Figure 1—figure supplement 4). Our results indicate that Myc‐expression is in fact heterogeneous. Furthermore, its expression does not correlate 100% with γH2AX staining, even though most of the γH2AX positive cells had Myc expression. We are not surprised by the heterogeneous nature of the Myc expression, given it is a protein with a short half‐life and the fact that the CMV promoter that controls Myc expression may not be universally active in all because it is subject to transient or long‐term silencing.

*3) Figure 1—figure supplement 1: there seems to be no difference in cell survival in either Casp3GFP high or low sorted cells. The authors should document by Annexin V (or other apoptotic markers) staining to verify whether apoptosis process is initiated in Casp3GFP high vs. low cells.*

The suggested experiment was done (Figure 1—figure supplement 2). Our results indicate that despite a significant increase in the fraction of cells with Casp3‐EGFP activation in Myc‐expressing MCF10A cells, the overall fraction of Annexin V positive cells did on increase at all.

*4) Do the authors find that knocking out Caspase 3 reduced the frequency of tumorigenic cells or is there an absolute prevention of tumorigenic growth? This aspect should be discussed. Further Caspase 3 may have other functions that are required for sustained tumor growth, such as suppressing MYC-mediated apoptosis. The authors need to acknowledge this possibility.*

Our answer to this important question would depend on the assays we used. If we used soft agar growth as the criteria, then the answer is that CASP3KO reduced the frequency of tumorigenic cells. If we use tumor growth in nude mice as the assay, then it is an absolute prevention of tumorigenic growth. We have now added some text on this point in the seventh paragraph of the Results and Discussion.

*5) The authors did not define the mechanism by which overexpressed MYC induces low level release of Casp3. This should be discussed to encourage further studies regarding MYC's effects on deregulated mitochondrial biogenesis and function.*

We have now added some language in the Results and Discussion section (seventeenth paragraph) about Myc’s potential effects on deregulated mitochondrial biogenesis and function as potential mechanism of Myc‐induced release of Casp3. References were also added for this point.